# Cross-Cultural Differences between Italian and UK Consumer Preferences for ‘Big Top’ Nectarines in Relation to Cold Storage

**DOI:** 10.3390/foods11162424

**Published:** 2022-08-12

**Authors:** Sarah R. Christofides, Anita Setarehnejad, Ruth Fairchild, Innocenzo Muzzalupo, Leonardo Bruno, Antonella Muto, Adriana Chiappetta, Maria B. Bitonti, Carsten T. Müller, Hilary J. Rogers, Natasha D. Spadafora

**Affiliations:** 1School of Biosciences, Cardiff University, Sir Martin Evans Building, Museum Avenue, Cardiff CF10 3AX, UK; 2Department of Healthcare and Food, School of Health & Sport Sciences, Cardiff Metropolitan University, Cardiff CF5 2YB, UK; 3Centro di Ricerca Foreste e Legno, Consiglio per la Ricerca in Agricoltura e L’analisi Dell’economia Agraria, 87036 Rende, Italy; 4Department of Ecology, University of Calabria, 87036 Rende, Italy; 5Department of Chemical, Pharmaceutical and Agricultural Sciences, University of Ferrara, 44121 Ferrara, Italy

**Keywords:** consumer survey, hedonic rating, nectarine, peach, *Prunus persica*, post-harvest, sensory attributes

## Abstract

Nectarines are perishable fruits grown in Southern Europe, valued for their sensorial properties. Chilling is used in the supply chain for Northern European consumers, while Southern European consumers can access fresh, locally grown fruit or cold-stored supermarket fruit. Cold storage and fruit ripening affect texture and flavour. Here a consumer survey and hedonic testing compared the appreciation of nectarines (cv. Big Top) in Italy and at two UK sites (n = 359). Fruit was at the commercial harvest stage, or stored at 1 °C or 5 °C for seven days, then sampled after two days’ (Italy and one UK site) or four days’ (second UK site) ambient recovery. In the consumer survey, the most important factors involved in purchase decision were ripeness, texture, colour, taste and price. Named varieties were more important to Italian than UK respondents, whilst ripeness, price, taste, blemishes, aroma, and ‘best before date’ were more important in the UK. In sensory analyses, fruits at the commercial harvest stage were preferred to those stored at 1 °C. Preference for the 5 °C stored peaches depended on recovery time. Distinct clusters of peach sensorial attributes were positively or negatively linked to hedonic rating. Factors important in purchase decisions did not affect hedonic rating in the tasting.

## 1. Introduction

Fresh fruit and vegetables are increasingly recognised as key components of a healthy diet [1,2] with protective effects against cardiovascular disease and colon cancer. Benefits are ascribed to intake of dietary fibre, vitamins, and phytochemicals, the latter including phenolics, vitamin C, and carotenoids in fruit. Fruits from the genus *Prunus*, which includes peaches and nectarines, are rich in phytochemicals [3] and hence are a useful contribution to a varied diet. Peaches and nectarines are a popular fruit in Europe with EU production at over three million tonnes in 2018; the second-largest EU producer is Italy, with over one million tonnes in 2020 [4]. However, peaches and nectarines are also widely consumed in northern European countries such as the UK, where the climate precludes production. For example, in 2019 the UK had a negative trade balance for fresh fruit of over four million euros [5]. Peaches are therefore transported long distances across Europe with at least two days’ transit time. Supply chains to supermarkets and other retail outlets then add extra days from producer to consumer. These fruits have a short shelf life, therefore chilled storage and transport is employed to delay ripening. However, major problems for the peach and nectarine industry include chilling injury and changes in texture and flavour during storage.

Chilling injury manifests as internal reddening and changes in texture, with a loss of flavour even before the appearance of visible symptoms [6]. This is most prevalent when fruit is stored or transported at higher temperatures of between 2–8 °C than at lower temperatures of −1 to +1 °C [7]. Food waste is of major concern, amounting in the EU to a yearly loss of 90 Mt [8]. This loss represents a challenge to food security, a waste of resources, and a source of greenhouse gas emissions from agricultural production and the supply chain [9]. In industrialised countries, household waste is a major contributor [10,11]. Fruits contribute over 80% of the wasted mass and over 40% of the wasted carbon footprint of supermarket food waste [9]. This is a particular problem for perishable fruit that travels long distances while chilled [12]. While increased control over transport conditions can reduce waste from fruit deterioration, concern about food freshness can increase wasteful behaviour by the consumer [13], as can fruit defects, both external and internal [14]. Thus, there is an important need to understand consumer assessment of fruit quality, to increase consumption while reducing waste.

Quality is a key driver for peach purchase and consumption [15], of which texture and flavour are key factors. Studies on peach purchasing in a number of different countries have identified key characters influencing purchase including firmness, colour, aroma, flavour, size, variety, sweetness, and lack of blemishes [16,17,18,19]. In different studies these quality attributes tended to group together, forming respondent classes or segments. Characteristics affecting purchase decisions can be divided into “experience” or “search” attributes [20], or may be “intrinsic”, “extrinsic”, or fall into the categories “experience” or “credence” [21]. In the context of fruit purchasing, experience attributes would include sweetness and flavour, which can be assessed only upon consumption, while firmness, size, and colour can be considered search or intrinsic qualities. Price and variety name are classed as extrinsic qualities, while healthiness is classed as a credence quality.

Ripening has a key role in improving peach organoleptic quality, including changes in pigmentation, increasing sugar to acid ratio, softening, and an increase in aroma [3]. Flavour is strongly influenced by the profile of volatile organic compounds (VOCs), constituting the aroma, and phytochemical content and VOCs can be assessed through biochemical approaches, e.g., [22]. However, it is essential to link sensorial analyses and chemical analyses with consumer perceptions [23]. Ripe soluble solids concentration was the major factor in overall appreciation when different peach and nectarine cultivars were compared [24], although other characteristics such as aroma and texture were also important. Indeed, firmness, colour, and aroma were the key characteristics in other studies [16]. Fruit maturity was a major factor in consumer preferences for ‘Big Top’ nectarines at harvest [25]. An analysis of the effects of maturity on key sensory characteristics [26] found that acidity, firmness, chewiness, and crispness fell with increased ripening, while pulpiness and juiciness increased. However, other important characteristics including odour and flavour intensity, sweetness, and fruitiness rose from unripe to semi-ripe fruit, but fell again in fully ripe fruit. Peaches are climacteric fruit in which ripening is coordinated by a burst of ethylene [27], and harvesting before this point results in impaired ripening. However, ripe fruit are more subject to damage and over-ripening before they reach the consumer. Storage temperature is also important in affecting peach sensory quality, with consumers showing a preference for storage at −1 °C compared to 4 °C [25], although no signs of chilling injury were noted even after seven weeks of storage.

Trends in food waste are similar across developed countries [28], although important differences have been noted in fruit consumption between southern and northern Europe [29]. Overall, more fruit consumption is found in southern compared to northern countries, as typified by the Mediterranean diet [30], despite a reduction even in southern countries [31]. Different dietary experiences across cultures may affect preferences for different sensory characteristics, but cross-cultural studies of food including fruit consumption have been relatively few [32]. Referring to peaches, Delgado et al. (2013) [24] studied different ethnic groups and found that in the USA white Caucasian subjects prioritised aroma, whereas for Asian and Asian American subjects sweetness was a key factor. However, there is no simple picture: for example, Spanish and Norwegian subjects with differing cultures but not ethnicity showed no difference in sweetness preference in apple juice [33]. There are still relatively few studies comparing sensorial appreciation of fruit across Europe. Given the cultural differences in consumption, with consequent potential effects on health, it seems timely to explore these differences to understand consumption drivers.

In this study, we compared preferences of the same nectarine cultivar “Big Top” which had been stored at different temperatures, at two stages of post-chilling recovery, between participants in southern Italy and the UK. Fruit tasting was carried out in university canteens to provide a food consumption context, which has been shown to be important for this type of study [34]. The aim of the study was to establish whether there were cross-cultural differences in consumer perception and purchase decisions.

## 2. Materials and Methods

### 2.1. Sample Material

The melting flesh, semi-free stone yellow peach cultivar (*Prunus persica* (L.) Batsch) ‘Big Top’ was used in this study. All peaches were within the FFV-26 recommendation for peaches and nectarines (Class I, size code 2A) [35] as stated by the provider “Campo Verde” Agricultural Company, Calabria, Italy. Peach samples were used at the commercial harvest stage without a cold storage period, or stored for seven days at 1 °C or 5 °C (D7-1C; D7-5C). They were then either kept at 22 °C for two days to reach the fully ripe stage (six boxes of circa 30 fruits) or sent to the UK via ambient road haulage in insulated boxes to maintain a temperature of 22 °C. Sampling was carried out in the 2019 summer season (10–12 July).

### 2.2. Sample Preparation

Samples were prepared for all tasting sessions in a food industry standard preparation area on the day of each consumer test. In Italy (site IT) stored samples were removed from cold storage and stored at 22 °C for two days prior to the consumer tests. On arrival in Cardiff (Wales, UK, following two days at 20–22 °C transit) the peaches were divided into two groups, for testing immediately at one site (Site CF1) and at the second site (Site CF2) after further room temperature (20 °C) storage for two days.

For each test site, on the day of tasting each peach was sliced to provide six segments of approximately equal size, avoiding the central stone (Figure 1), by researchers wearing appropriate personal protective equipment for a food preparation area. Samples were provided to participants in transparent polypropylene cups of 100 mL capacity, each sample type randomly coded with a 3-digit code to reduce expectation error [36].

### 2.3. Consumer Test

When all the samples for each day of consumer testing were ready, they were transported in insulated food-grade boxes to the test sites. Products were served at room temperature of approximately 20 °C. Fruit for the site CF2 consumer test were left whole and unboxed on the tables in the food preparation area, clearly labelled with their codes. Site CF2 peaches were served as for site CF1 and site IT outlined above.

Ethical approval for the study was obtained from the appropriate committees (Sta-1353 and SREC 1906-02) before commencement of the consumer test. The test was set up in three locations over the course of two days. On day one of the study, testing was undertaken in the main catering facility of a university situated in southeast Wales, UK (site CF1) and in the main hall of a university campus canteen in Calabria, Italy (site IT). On day three it took place in the main foyer of a second university in southeast Wales, UK (site CF2).

The consumer test followed BS EN ISO (2020) [36] guidance for conducting hedonic tests with consumers in a controlled area, with controlled preparation and presentation of the products and comfortable conditions for consuming the products and for questioning the consumers.

The test consisted of a printed questionnaire (Appendix A) of closed questions [36] administered by the researchers and additional postgraduate students, who invited participants (university staff, students, and visitors) in each location to take part in the tasting. The questionnaire was translated by one native Italian researcher for use at the Italian site (Appendix A). Tests were conducted within the lunch period of 12.00–15.00 to maximise the number of participants. Completion of the questionnaire took around 5–10 min. Participants were first informed about the purpose of the study, to gain their assent to take part, then they were provided with the questionnaire, a pen, and three randomly coded ‘Big Top’ peach samples. Attributes addressed in each question were selected based on a review of existing literature concerning sensory aspects of various fresh fruit including peaches [37,38,39,40], and purchase decisions with regard to fresh fruit [19,41]. Geo-demographic information (personal characteristics) was entered at the start of the questionnaire (age, gender, ethnicity, nationality, site of study, and frequency of eating peaches). Participants were also asked whether they sought named varieties when selecting peaches. This was followed by a series of questions to determine the decision attributes of importance to the participant when selecting fresh peaches (peach characteristics: ripeness, colour, size, shape, lack of blemishes, price, aroma, texture when handled, taste, retailer, best before date, and other).

Participants then tasted the samples provided, in a forced randomised order indicated on the questionnaire to reduce positional bias or order effect [36], and rated the samples using a nine point Likert rating scale with verbal labels of extremely dislike = 1 and extremely like = 9. Likert scales are frequently used in sensory studies concerning purchasing, attitudes, and sentiments, including preference [42]. The questionnaire concluded by asking participants to circle the descriptors which best matched each sample cultivar, from a list derived from literature [37,38,39,40]: peachy aroma, sweet, acid, bitter, astringent, juicy, crunchy, chewy-fibrous skin, firm flesh, soft flesh. At the end of the questionnaire, participants were reminded that return indicated their consent for data to be used for analysis. As the questionnaires were anonymous, participants were made aware their responses could not be removed after submission of the questionnaire.

### 2.4. Statistical Analysis

Statistical analysis was conducted in R v3.6.3 (R Foundation for Statistical Computing, Vienna, Austria) using RStudio and packages car, dplyr, dunn.test, emmeans, gridExtra, ggmosaic, ggplot2, igraph, mvabund, multcomp, MuMIn, ordinal, RVAideMemoire, tidyr, tools, and vegan [43,44].

Using a binomial multivariate generalized linear model (binomial manyglm), relationships were modelled between the characteristics influencing the decision to purchase peaches, frequency of purchasing peaches, and national effects. Univariate tests were conducted using analysis of deviance with a step-down resampling procedure for *p* value adjustment. A separate model checked for any effect of participant gender. The relationships between hedonic responses (liking), site, treatment, and the site–treatment interaction were modelled using a cumulative link mixed model for ordinal response variables. Respondent ID was included as a random term to account for non-independence. Pairwise tests were conducted based on least-squared means with Tukey *p* value adjustment.

A model averaging approach was used to assess the influence of (a) peach characteristics and (b) respondent characteristics on the hedonic responses (liking). Cumulative link mixed models were used due to the ordinal response variable. Respondent ID was a random term in all models. Two groups of models were constructed: peach characteristics in all possible combinations and personal characteristics in all possible combinations. An Akaike information criterion test (AICc) was used to rank models within each group, from best (lowest AICc) to worst (highest AICc). Within each group, models with ΔAICc < 6 (i.e., up to six times the AICc of the best model in the group) were considered good [45]. For the peach-based models, a further subset was taken where ΔAICc < 2, i.e., up to twice the AICc of the best model in the group, which was considered indistinguishable from the best model [45]. Coefficients from these models were averaged to obtain robust estimates of the effect of each predictor.

The co-occurrence of pairs of peach characteristics was tested using Fisher’s exact tests. *p* values were adjusted for multiple testing using the Benjamini–Hochberg procedure [46]. Significantly co-occurring pairs were visualised as a network. The effects on characteristic scores of site, treatment, and site–treatment interaction were modelled using a binomial multivariate generalized linear model. Univariate tests were conducted using analysis of deviance with a step-down resampling procedure for *p* value adjustment.

Finally, for every sensorial characteristic, the mean hedonic rating was calculated per respondent for (a) samples where that characteristic was reported present, and (b) samples where it was reported absent. The difference in hedonic ratings between the two cases was regressed against the respondent’s gender, country, frequency of purchasing peaches, and whether they considered particular characteristics important when purchasing peaches. *p* values were adjusted for multiple testing using the Benjamini–Hochberg procedure [46].

### 2.5. Strengths and Limitations of the Study

The selected test methodology is suitable for studying the impact of sensory characteristics of a product on degree of liking, independently of the product’s extrinsic characteristics such as brand, and studying the effect of a commercial or presentation variable, in this case transport [36]. The method is effective for determining whether or not a perceptible preference exists between products [36].

Test Environment: It was not possible to control completely the absence of communication (verbal and non-verbal) between the consumers, which would guarantee independent responses. Researchers did ask the consumers not to discuss the results with each other and were nearby to remind consumers of this during the test. Minimal interaction was observed.

Test timing: The time of a test can influence its results. Sensory guidelines indicate that 10 a.m.–lunch is a suitable time for tasting as it corresponds to usual consumption time for products such as fruit, e.g., as a part of lunch or a snack. In our study, the timing was over the lunch period. However, we did ask participants to consume the peaches before rather than after their lunch, to minimise the impact of sensory overload or taste masking [35] (BS EN ISO 2020)

Questionnaire wording: It is important to ensure that cultural differences are considered in the development of questionnaires. A particular problem was translation of the questionnaire [32]. Here the study team included native English speaking and native Italian speaking food-research specialists, as well as two bilingual authors to ensure that all terms used were as far as possible equivalent. Despite this, data were collected on two additional descriptors (“fragrante” in Italian and “mealy/mushy flesh” in English), but these were excluded from the final analysis since they represented a language barrier. In addition, “consistente” was merged with “dura” by the Italian team, as these terms constituted linguistic ambiguity.

Profile of participants: The experiment was conducted in a university setting which might bias the respondents towards young people (students) with lower experience and income levels but higher education levels (compared to the population). However, the experiment was conducted outside term time and thus mainly included University staff and postgraduate students. In addition, this applied to all three sites, making them comparable. However, the samples may still not be representative of the whole populations in the two countries in terms of age and educational level.

## 3. Results

### 3.1. Participant Information

A total of 359 participants returned completed assessments for analysis across the three sites, 103 participants (28.69%) from site CF1, 101 (28.13%) from site CF2 and 155 (43.18%) from site IT. The total sample consisted of 174 males (48.47%), 179 females (49.86%) and six who did not disclose their gender (1.67%). Of the 359 participants, 165 (45.96%) were aged 18–29 years old whilst only 18 (5.01%) were over the age of 60, with nearly equal distribution of the remainder between the 30–39 (61; 16.99%), 40–49 (56; 15.60%), and 50–59 (59; 16.43%) age groups (Appendix A).

The majority of participants described their ethnicity as Caucasian (n = 302, 84.12%), with less than 5% in each category of black (17; 4.74%), Asian (16; 4.46%), Hispanic (13; 3.62%); other was used as an ethnicity description by only 1.95% (n = 7) of the participants, and 1.11% (n = 4) did not answer the question (Appendix A). A total of 157 participants (43.45%) described their nationality as British (including Welsh, English, Scottish) and 116 as Italian (32.31%). Other nationalities included various European (Dutch, Polish, EU, etc.), African (Chad, Somalia, Moroccan), Indian, Pakistani, Chinese, Australian, and American (Appendix A).

### 3.2. Characteristics Influencing Purchase Decisions

Across all participants, 37% purchased peaches less than once per month, 23% weekly to monthly, and 40% more than once per week. However, buying frequency varied significantly with country (χ^2^ = 84.3, *p* < 0.001), and most (73%) of the Italian respondents were in the frequent buyer category (>1 per week), while over 50% of UK buyers bought peaches less than once per month (Figure 2A).

The most important factors in the decision process of selecting peaches across all respondents were ripeness, texture, and colour (Figure 2B), each selected by over 1/3 of respondents. There was no effect of participant gender (manyglm, *p* > 0.05; Appendix A). Univariate tests revealed differences between Italy and the UK in the importance of ripeness, price, taste, variety name, blemishes, aroma, and best before (BB4) date (Figure 2B; Appendix A). In all of these cases apart from variety name, UK testers were more likely than Italian respondents to consider the named factors important (Figure 2B). Indeed, the combination of factors influencing decision to purchase differed between countries (LR_1,310_ = 203.4, *p* = 0.001; Figure 2B). However, once national effects were controlled for, in terms of determining which peach characteristics were considered important, there were no purchase differences between the frequency categories nor any significant interaction between country and frequency of purchase (Appendix A).

### 3.3. Differences in Hedonic Rating Relating to Peach Storage and Site

There was significant interaction between site and treatment in predicting hedonic rating scores (LR_4,18_ = 31.6, *p* < 0.001). At site IT, scores among the three treatments were remarkably consistent (Figure 3); the only significant difference was a preference for commercial harvest peaches over those stored at 1 °C (*z* = 3.52, *p* = 0.013). Likewise at site CF1, testers significantly preferred the commercial harvest peaches over those stored at 1 °C (*z* = 6.07, *p* < 0.001), and also preferred them over those stored at 5 °C (*z* = 4.80, *p* < 0.001). At site CF2, in contrast, 5 °C stored peaches were significantly preferred to 1 °C stored peaches (*z* = 5.15, *p* < 0.001), but commercial harvest peaches were not significantly preferred to either storage treatment.

There was no significant difference in preference for the three treatments between sites IT and CF2, even though the site CF2 tasting took place two days later than those at sites IT and CF1. Hedonic ratings at sites IT and CF1 were significantly different for both storage treatments (1 °C: *z* = 6.40, *p* < 0.001; 5 °C: *z* = 5.70, *p* < 0.001), but only marginally significantly different for the commercial harvest peaches (*z* = 3.12, *p* = 0.047). Sites CF1 and CF2 differed significantly only for the 5 °C storage treatment (1 °C: *z* = 6.50, *p* < 0.001).

### 3.4. Fruit-Related Attributes Influence Hedonic Rating More Than Person-Related Characteristics

Two groups of models were constructed to explain hedonic rating: peach attributes (aroma, sweetness, acidity, bitterness, astringency, juiciness, crunchiness, fibrosity, firmness, and softness) in all possible combinations, and person characteristics (site, gender, age, ethnicity, and frequency of purchase) in all possible combinations. AICc was calculated for each model to rank models within each group from best (lowest AICc) to worst (highest AICc). The peach-based models had consistently lower AICc than the person-based models (Figure 4A), indicating better explanatory power.

Within each group, models with a ΔAICc < 6 (i.e., up to six times the AICc of the best model in the group) were considered good [45]. The numbers of times each predictor occurred in these sets were counted (Figure 4B,C). Sweetness, juiciness, aroma, and bitterness appeared most frequently in the peach-based models, whilst site was the most frequent in the person-based models.

For the peach-based models, a further subset was taken where ΔAICc < 2, i.e., up to twice the AICc of the best model in the group, which was considered indistinguishable from the best model [45] (Appendix A). Coefficients from these models were averaged to obtain robust estimates of the effect of each predictor (Figure 5). Aroma, juiciness, and sweetness were consistently positively associated with liking, whilst astringency, bitterness, and crunchiness had consistent negative effects.

### 3.5. Co-Occurrence of Peach Characteristics

Of the forty-five pairwise combinations of characteristics, twenty-seven were significantly non-randomly associated with each other (Appendix A). Figure 6 shows the strength and direction of these significant combinations. The three characteristics associated with increased hedonic rating (juiciness, sweetness, and aroma) were strongly positively linked with each other and with softness, when all respondents were taken together. However, the links differed in strength between sites (Appendix A), for example, sweetness and juiciness were strongly linked at the two UK sites but only weakly linked at the IT site. Crunchiness was positively associated with firmness at the two UK sites but not at the IT site. Among all respondents, bitterness and acidity were positively associated with each other but not to any other characteristics, and astringency had only negative associations. This indicates that good peaches share the same essential characteristics, but that multiple attribute groupings could define bad peaches, some of them with mutually exclusivity.

### 3.6. Effect of Site and Treatment on Perception of Peach Characteristics

Overall, there was significant interaction between site and treatment affecting the perception of peach characteristics (LR_4,1068_ = 184.7, *p* = 0.001). Taking the characteristics individually, acidity, aroma, crunchiness, firmness, juiciness, and sweetness were significantly affected by the interaction between site and treatment (Figure 7; Appendix A). Acidity at IT and CF1 was rated least in the commercial harvest stage fruit, while aroma and sweetness were rated highest in the commercial harvest stage fruit at both these sites. Juiciness, crunchiness, and firmness showed varying trends across sites and treatments. Bitterness and astringency were significantly affected only by treatment, and commercial harvest stage peaches were assessed as less bitter than stored peaches. Fibrosity was significantly affected only by site, and was higher at CF2 than the other two sites. Softness was significantly affected by site and treatment but with no interaction, and was rated highest at CF2 and for peaches stored at 5 °C (Figure 7; Appendix A).

### 3.7. Relationship between the Stated Importance of Characteristics and Their Influence on Hedonic Rating

Aroma was the only characteristic that could be assessed from questionnaire responses as a driver for purchase and in the hedonic tasting as a factor related to liking. Respondents that rated aroma as an important factor in purchasing peaches did rate aromatic peaches as slightly more liked than purchasers who did not consider aroma important for purchase decisions (Figure 8). However, there was much more variability amongst the latter group of respondents, and the difference in rating was not statistically significant.

The influence of all the other sensorial characteristics on hedonic rating was likewise remarkably consistent among consumers, with no significant differences based on country, gender, frequency of purchase, or which characteristics were rated as important in purchase decisions (Appendix A).

## 4. Discussion

### 4.1. Characteristics Considered Important in Purchase Decisions Differ between Countries

As recently noted, fresh goods buyers’ purchasing behaviour follows the same established patterns as observed in consumers buying packed goods [47], and less frequent buyers contribute more to purchasing than frequent buyers. This trend was supported in the present study (60% of respondents bought peaches once a week or less) with many more infrequent or light buyers than heavy ‘lovers’ [47]. Therefore, of particular interest is that these groups did not differ in which characteristics they most appreciated, despite substantial differences between Italian and UK respondents. It has been previously noted [32] that southern European respondents can be more acquiescent than northern European respondents; this was not seen here (Kruskal–Wallis χ^2^ = 0.907, *p* = 0.34), suggesting that this does not explain the cross-cultural differences noted. However, we showed that there was a significant difference in the factors influencing purchasing decisions between respondents in the two countries, and that the country of the respondent was more important than frequency of buying in influencing the importance of different purchase characteristics.

Of the twelve characters relating to purchasing decisions included here, four can be considered extrinsic characteristics (price, variety name, best before, and retailer), five intrinsic (shape, size, aroma, blemishes, and colour) and three relate to experience (ripeness, texture, and taste) [21]. The findings that ripeness, texture, and colour are the most important characteristics to all groups of purchasers broadly fits with other surveys of peach-purchasing drivers [16,17,18]. Texture was rated highly in both countries, an important characteristic highlighted by other studies [16,18,19]. Ripeness is unequivocally related to texture, with increasingly softer and more juicy flesh [48], and is generally appreciated by consumers.

One key difference between UK and Italian respondents was that those in Italy were more likely to consider variety name as important when purchasing, suggesting that peach varieties are not broadly known by consumers in the UK. Few studies have considered variety name as a factor in fruit purchase decisions. Selection by variety is very important in apples, where it has been linked to high frequency purchasers [49], however, in other fruit, such as kiwis, novelty in varieties may be rejected by sectors of consumers [50]. The response from participants at the Italian site relating to variety name as a factor may be linked to a greater familiarity with peach and nectarine varieties, as Italy is a major peach-producing country. This would be consistent with Canadian peach purchase trends where regional labelling was a relevant purchasing factor [18], as Canada also has peach-producing regions. Other attributes, extrinsic and intrinsic as well as experience-based, were rated more highly by UK respondents, perhaps reflecting a more considered approach to peach purchase compared to Italian respondents who were more frequent buyers.

The presence of blemishes was more important for respondents in the UK than in Italy. The high proportion of buyers who place a high value on unblemished fruit has been noted in previous studies [14,19,51]. This characteristic may also relate to credence qualities [21], i.e., healthiness. However, the best-recognised extrinsic quality indicator, price, was also much more important to UK respondents. In the UK, one survey indicated that price was key factor in the purchase of fresh produce for the majority of consumers [52], and more recent studies suggest that price remains a barrier to purchase [53,54]. In contrast, price was an important factor for about half of German peach buyers [55]. For UK respondents, the best before date was of relevance, as has been found previously in Spain [56]. Experience qualities such as taste were also important, as has also been noted in previous studies where it was linked to other experience attributes such as sweetness [19] contributing to a class of “experience attribute-oriented consumers”.

### 4.2. Hedonic Rating Reflects Peach Freshness and Ripening

Trained panels are frequently used in the sensory evaluation of fruit, including peaches e.g., [25,57]. However, this method does not reflect true consumption patterns. Our study used hedonic testing with untrained consumers in a controlled area, which better represents consumers’ eating behaviour [36], albeit perhaps not representing whole population demographics (see Section 2.5) and ‘strong feelings’ due to cultural differences in food consumption habits [58]. Culture and familiarity with a food product may influence overall consumer perception of food and drinks [59]. Given that Italy is a major peach grower, which the UK is not, peaches are likely to be more familiar and culturally important to Italian consumers. This may explain the generally higher hedonic ratings from Italian respondents. One strength of our study is the substantial participant numbers (n = 359), similar to a recent cross-cultural study on consumer preferences for apple juice [33] in Norway and Spain, which utilised 125 participants in each location. Moreover, sensory trials have often been dominated by female respondents [32], due to their location and emphasis on food selection. However, in our study there was no significant difference in gender distribution (48% male and 50% female). The age of respondents in our study reflected its occurrence on university premises at all three sites, thus nearly half (45.9%) were aged 18–29 years and only 5% aged over 60 years. This concentration of younger age groups has been found in other research originating in university settings, including studies examining peach preferences [41] and pear consumer preference mapping [60]. Sensory discrimination is known to reduce from the age of 60 [61] so a largely young participant base could be considered a benefit in consumer-focused studies (with the demographic limitation noted above). In addition, purchase behaviours in general, and of fruit such as apples in particular, have been difficult to identify using demographic information such as age [49].

Comparing the two sites where peaches of equivalent ripeness were tasted (sites IT and CF1), it was found that commercial harvest stage peach samples were preferred at both sites. This perhaps suggests that attributes reported as related to freshness, such as taste, crispiness, juiciness, and flavour [40], were detected more in the commercial harvest stage peaches than in the cold-stored samples. However, at site CF2, where the fruit presented were stored for an extra two days at ambient temperature, the difference between commercial harvest stage and cold-stored fruit was no longer significant. This probably reflects that increased ripeness is a key factor in peach appreciation, as previously noted [25]. The sample stored at 1 °C remained the least preferred, which could be due to the effect of low temperature delaying the ripening of the fruit. Lowering the storage temperature slows metabolism [62], and recent modelling [63] shows that the development of sweetness index is substantially reduced by cold storage.

Cold storage is known to affect hedonic qualities in peaches, although these are not always detectable after only one week in storage [25,57]. At all three sites, bitterness was lower in commercial harvest stage fruit compared to fruit stored at 1 °C. This could be due to delayed ripening at the colder temperature, previously detected as increased acidity in colder stored fruit [57]. The ripening effect was indeed most clearly seen at the two sampling sites that received the less ripe fruit (IT and CF1), where the peaches stored at 1 °C were rated most acid, least sweet, and with least aroma. The reduction in sweetness was perhaps somewhat unexpected, however, because other studies have not detected expected changes in sweetness during storage [25]. This may relate to the acid–sugar balance which is critical in peach appreciation [41].

### 4.3. Influences on Hedonic Rating Are Consistent among Participant Groups

Overall, the characteristics of peaches were more important compared to person-related characteristics for determining consumer appreciation at tasting, indicating a general agreement across respondents on the preferred characteristics in peaches. Amongst the person-related characteristics, site was the most important factor, perhaps due to the differences in peach ripeness between sites CF1 and CF2. In agreement with previous studies [24,26], sweetness, aroma, and juiciness were consistently liked. In this study, astringency was consistently disliked; this contrasts with data from Predieri et al. (2006) [64], whose study also included ‘Big Top’ nectarines and who found an association between astringency and liking. Several studies have shown variability in the key attributes linked to liking, and Delgado et al. (2017) [24] in particular linked this to ethnicity, although this was not tested in the present study. However, in our study the peach attributes also correlated with each other. The association of juiciness, sweetness, aroma, and softness were likely due to ripening effects [3,26], as was the link between bitterness and acidity. The association of crunchiness with firmness was also consistent with the changes in texture seen during peach ripening. There were three distinct groups of negatively associated characteristics (acidity/bitterness, firmness/crunchiness, and astringency) compared to just one group of positively associated characters (juiciness/sweetness/aroma/softness), which may indicate the effects of the different treatments.

An important finding was that respondents’ stated preference for particular characteristics when purchasing peaches had no significant effect on their liking for the samples once tasted. For example, it might have been predicted that respondents who listed aroma as an important purchase criterion would more heavily penalise peach samples that they did not perceive as aromatic. The absence of such an effect suggests that there may be a disconnect between what a consumer expects to like and what they actually enjoy. This has implications for the supply chain, as the criteria used by a consumer at the point of purchase need to be differentiated from their perceptions at the point of consumption.

## 5. Conclusions

Storage of peaches at higher temperatures has been shown to improve consumer appreciation [25] but is known to increase spoilage and chilling injury [7] with consequent waste. These effects vary across cultivars, but data from this study shows that UK consumers may be less aware of cultivar differences. Moreover, there was a link between respondents in Italy, who were also more frequent buyers, and an understanding of peach varieties. Peaches at the commercial harvest stage were preferred to those stored at 1 °C, while peaches stored at 5 °C were rated more favourably with increasing recovery time, suggesting that ripeness was the driving factor. Peach sensorial attributes formed distinct clusters; good peaches were juicy, sweet, aromatic, and soft, while bad peaches could be acid, bitter, firm, crunchy, or astringent. Respondents showed remarkable consistency in their appreciation of different peach attributes, irrespective of demographic differences and preferred purchase criteria. Improved information on cultivar resilience to chilled transport, increased education of northern European consumers on varietal differences, and feedback to breeders on the qualities most appreciated may be combined by the industry to reduce peach spoilage and waste.

## Figures and Tables

**Figure 1 foods-11-02424-f001:**
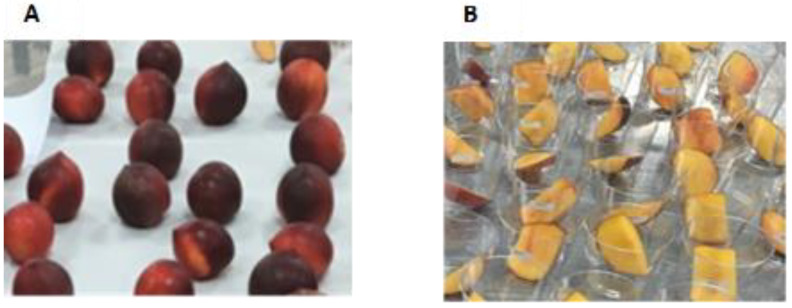
‘Big Top’ peach samples use for sensorial analysis (**A**) samples on unpacking, (**B**) samples as prepared for tasting at Cardiff sites.

**Figure 2 foods-11-02424-f002:**
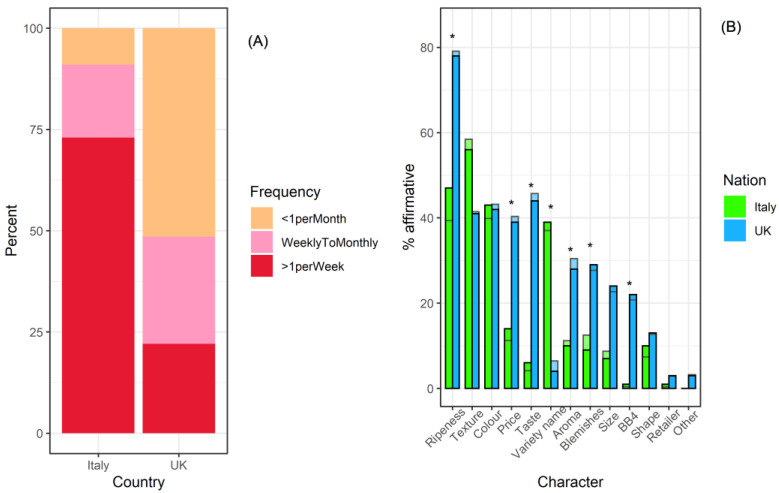
Importance of peach characteristics influencing purchase decisions. (**A**) Distribution of frequency-of-purchase classes between countries. (**B**) Analysis based on the percentage of respondents who said that the characteristic was important to them, grouped by country. Opaque bars with black outlines represent the actual data; semi-transparent bars with grey outlines represent model predictions controlling for other variable (i.e., predicted value for each country whilst controlling for frequency of purchase, and vice versa). Significant differences (*p* (adj) < 0.05) are indicated with a star (*). BB4 represents the best before date.

**Figure 3 foods-11-02424-f003:**
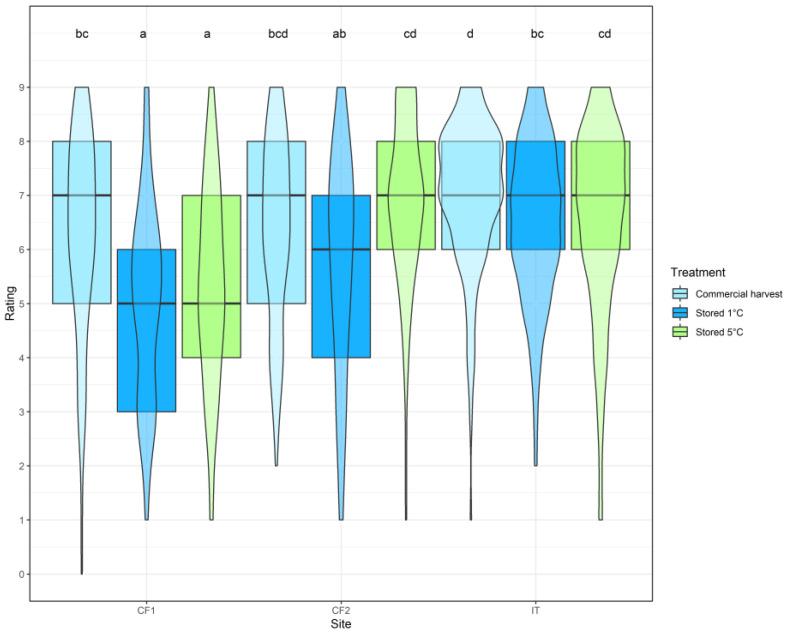
Hedonic responses to three samples of ‘Big Top’ peach stored under different conditions tasted at three locations (Italy and UK). Participants were asked to rate peaches on a 9-point scale. Boxplots show the median and quartiles for each group, while the violin plots indicate the proportion of data points at each value. Letters indicate significant differences.

**Figure 4 foods-11-02424-f004:**
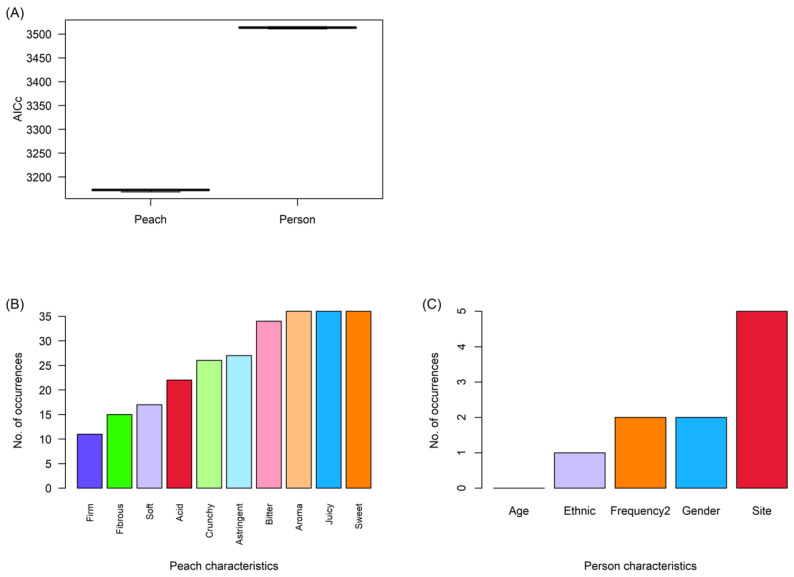
Importance of peach attributes and person characteristics in determining the ‘liking’ rating given to a peach, based on ranking model sets by AICc. All models with a delta value < 6 (acceptable) are included here. (**A**) Boxplots of AICc for peach-based vs. person-based models. (**B**) Number of times each peach attribute appeared in the reduced model set. (**C**) Number of times each person characteristic appeared in the reduced model set.

**Figure 5 foods-11-02424-f005:**
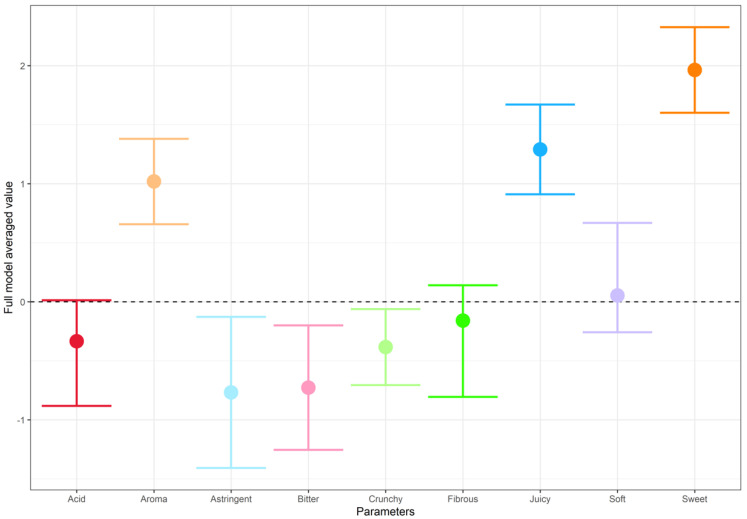
Model averaged parameter values for peach characteristics, based on only the best models (delta value < 2). The parameter value represents the average change in rating based on that parameter moving from ‘No’ to ‘Yes’, the dashed line indicates no effect. Thus, for example, sweetness had a consistently positive effect on hedonic rating whereas bitterness had a consistently negative effect.

**Figure 6 foods-11-02424-f006:**
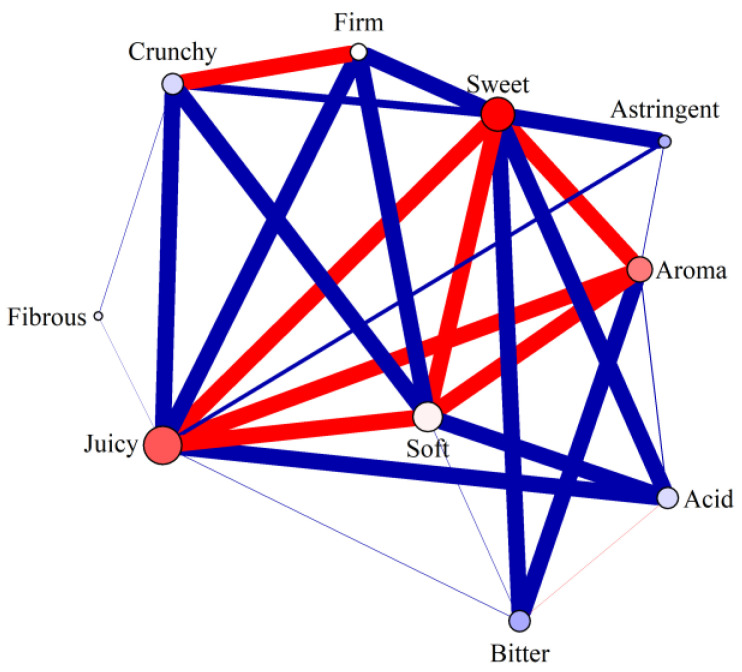
Co-occurrence frequency of peach characteristics. Only significant associations are shown; for a full list see Appendix A. Edge widths are inversely proportional to *p* values (thick edges indicate low *p* values). Red edges indicate that a pair of nodes co-occurred more often than expected by chance; blue edges indicate they co-occurred less often than expected by chance. Node colour for each character is proportional to the corresponding parameter value in Figure 5: blue indicates a negative effect on hedonic rating and red a positive effect, while the strength of colour corresponds to the strength of the effect.

**Figure 7 foods-11-02424-f007:**
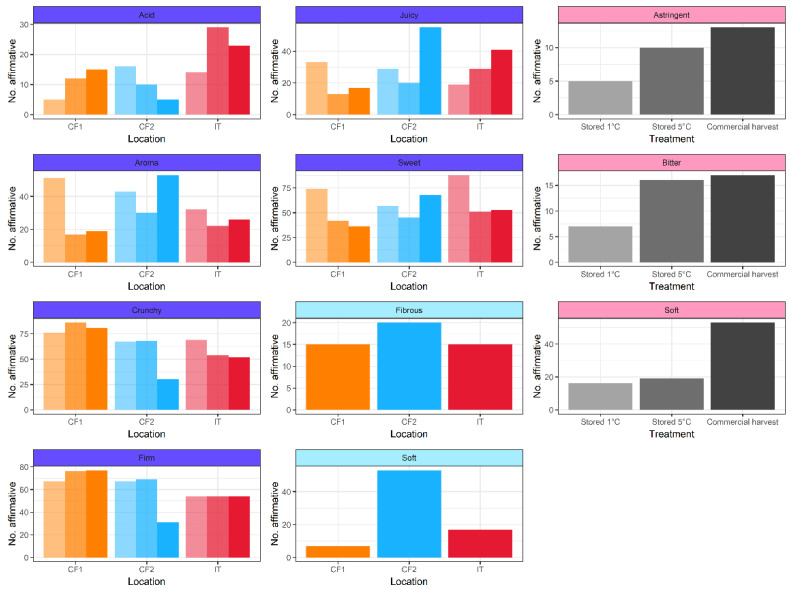
Influence of treatment and location on perceptions of peach characteristics. Sites are coded by colour of the bars (orange = IT, blue = CF1, red = CF2); treatments are coded by shading (light = seven days’ storage at 1 °C, intermediate = seven days’ storage at 5 °C; dark = commercial harvest stage). The colour of the facet bar indicates the result of the manyglm model: purple = significant interaction between site and treatment, blue = significant effect of site, pink = significant effect of treatment.

**Figure 8 foods-11-02424-f008:**
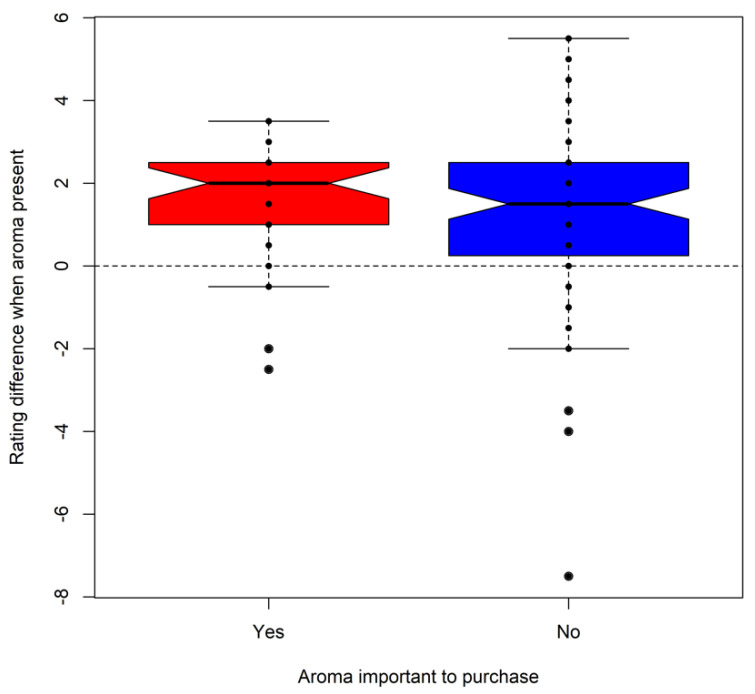
The mean difference in hedonic rating when respondents regarded a peach as aromatic. Data are plotted against whether that respondent rated aroma as an important factor influencing their purchase decisions.

## Data Availability

Data available upon request.

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
