# Peer review of "Cross-Cultural Differences between Italian and UK Consumer Preferences for ‘Big Top’ Nectarines in Relation to Cold Storage"

_foods, 2022, doi:10.3390/foods11162424_

Round 1

Reviewer 1 Report

The authors investigated the effects of different storage temperatures on the liking of ‘Big Top’ peach samples at three sites in two countries (Italy and UK). The frequency of consumption and familiarity with the fruit/variety were not the same between consumers from the two studied countries, and “site” and “storage temperature/duration” factors were identified as major contributors in determining liking ratings.

While the study appears to be sound, it fails to address three major concerns listed below:

1.    Conducting consumer research studies using convenience sampling at university campuses is the significant limitation of the study. Young university students with lower experience and income levels but higher education levels (compared to the population) cannot be the representative of their population in questions related to ‘liking’ a product or attributes considered in making a purchase decision. Therefore, the results of the study cannot be generalized to the consumers of the studied countries. The statements in “L 512-521” indicate that the authors considered this limitation as a positive contributor in the validity of their study while it should have been presented clearly as a limitation of the study that impacts the generalizability of the results if the data have not been weighted considering the census demographic data from each country.

2.    The authors should clearly discuss why their study is considered a “cross-cultural” study if the majority of the population in both countries are from the same ethnic background and how the effects of the confounding factors (i.e., greater familiarity of Italian consumers with the product and their higher consumption frequencies) were addressed in their study?

3.    In the current presentation format, results demonstrate a study with a full-factorial design (3 x 3). Nevertheless, “CF2” samples were not only different in the location of the study but also in their storage duration. Thus, the effects of the location (i.e., “site’) and storage cannot be discussed independent from each other for “CF2” samples, and the authors should analyze data, present results and interpret their findings considering this important fact.

Minor issues are listed below:

-       L 123: Please include information about the relative humidity of the storage rooms in addition to their temperature.

-       L 170: “closed-ended” questions not “closed questions”

-       L 174: It is not recommended to conduct sensory/consumer research tests with tasting sessions scheduled for lunch time. Please elaborate on the potential impact of this factor on the validity of the results.

-       L 201: Please use the common names of the statistical tests applied rather the names of R-packages. For example, the “binomial manyglm” refers to “High Dimensional Generalized Linear Models”.

-       L 239: A table is required to show the demographic characteristics of respondents per site, and the demographic characteristics of UK and Italy populations, in separate columns. Presenting them in the current format is confusing.

-       L 285-286: Is the term of “commercial harvest peaches” refer to “fresh” peaches? The treatment levels should be addressed using the same names/terms consistently in the manuscript.

-       L 326: “Letters indicate significant differences.”: Please elaborate why there are two significance levels of “ac” and why the significance level of “b” is missing from “ac” levels.

-       L 445: “grey = no significant differences”; which graphs is this statement referring to?

-       L 459: “than” instead of “that”

-       L 479: Is the “variety” word used to refer to the “variety name”?

-       L 573: Is the “chilling injury” a problem resulted by “higher temperatures”?

Reviewer 2 Report

In the manuscript: Cross cultural differences between Italian and UK consumer preference for „Big Top” nectarines in relation to cold storage, the Authors tried to compare preferences of the same nectarine cultivar “Big Top” which had been stored at different temperatures, at two stages of post-chilling recovery, and between participants in Southern Italy and the UK. The Authors wanted to assess cross cultural differences in consumer perception and purchase decisions. 

Generally, the manuscript provides valuable information. The Authors clearly described sample material and preparation. The consumer test was conducted properly with the guidance for conducting hedonic tests with consumers in a controlled area. The Authors got all approval from ethics committee. Additionally, the Authors used proper tests for the data in statistical analysis. However, I have some remark.

The Authors wrote that only 5.01% of participants were over the age of 60. Have you not thought about excluding these people from the study? This disproportion may have affected the results.

I suggest a separate section for the strengths and limitations of the study, making it easier for the reader to perceive.

Round 2

Reviewer 1 Report

Thank you for responding to my questions/comments, and for applying the revisions. I think the first major issue is addressed properly by the authors by limiting the results to the respondents rather than the populations of the two countries. However, I think the manuscript can still be improved further and I have listed a few comments and resources below (related to the initial review questions).

1.    The second major issue (i.e., if the study is a “cross-cultural” study) needs to be discussed in more details. Rødbotten et al.’s (2009) study was different from the presented one as Rødbotten et al. studied a product (i.e. apple juice) that was similarly available in both countries they conducted their study. The emphasize in the current study should be on basing the differences in food preferences on differences in “familiarity” of the respondents with the product (i.e., peach) considering their cultural background, and then investigating their preferences in regard to the storage temperatures and conditions. The influence of “familiarity” through exposure on consumers’ unconscious (and conscious) food choices have been explored in different studies (some listed below). Foods that are culturally embedded in one’s culture are learned and can therefore reinforce preference, and the authors need to clearly discuss this and cite references similar to the listed ones below:

·         Jeong, S., & Lee, J. (2021). Effects of cultural background on consumer perception and acceptability of foods and drinks: a review of latest cross-cultural studies. Current Opinion in Food Science42, 248-256.

·         Köster, E. P. (2009). Diversity in the determinants of food choice: A psychological perspective. Food quality and preference, 20(2), 70-82.

·         Li, M., and Chung, S. J. (2021). Flavor principle as an implicit frame: its effect on the acceptance of instant noodles in a cross-cultural context. Food Quality and Preference, 93, 104293.

·         Prescott, J. (2020). Development of Food Preferences. Handbook of Eating and Drinking: Interdisciplinary Perspectives, 199-217.

·         Rozin, E. and Rozin, P. (1981). Culinary themes and variations. Natural History, 90, 6–14

·         Sato, W., Rymarczyk, K., Minemoto, K., Wojciechowski, J., and Hyniewska, S. (2019). Cultural moderation of unconscious hedonic responses to food. Nutrients, 11(11), 2832.

·         Stallberg-White, C., and Pliner, P. (1999). The effect of flavor principles on willingness to taste novel foods. Appetite, 33(2), 209-221.

·         Sulmont-Rossé, C., Drabek, R., Almli, V. L., van Zyl, H., Silva, A. P., Kern, M., ... and Ares, G. (2019). A cross-cultural perspective on feeling good in the context of foods and beverages. Food research international115, 292-301.

2.    The fact is that the samples tested in “CF2” were different from those tested in the other two sites (because of different storage durations). Therefore, three sites should not be compared directly. In agricultural and food science studies the best way to address these types of experiment designs is to use nested designs rather than full-factorial experiment designs. The examples referred to in the response document (“lines 527-248”) were not clear to me. Nevertheless, I am not requesting any revisions on this issue at this stage anymore.

3.    Table S1 can be useful if presented in percentages alongside the census data for the two countries. In addition, considering the very few participants from the listed nationalities except British and Italian, there is no need for presenting the detailed nationality data and the ‘other’ category can be used to show the number of participants who were not British or Italian.

All the best,
